# Learning Gaussian Mixture-distributed Prototypes for 3D Scene Graph Generation from RGB-D Sequences

Rongxing Ding [* 1]   Hongyu Qu [* 1]   Xinguang Xiang [1]   Pengpeng Li [1]   Xiangbo Shu [1]

## Abstract

3D Scene Graph Generation (3DSGG) aims to create a structured representation of 3D environment by identifying objects as nodes and their relations as edges. Existing 3DSGG methods based on RGB-D sequences typically put much focus on the adaptation of neural networks to robust node and edge feature extraction in complex 3D scenes, while ignoring the inherent intra-class diversity within each class and inter-class similarity between different categories associated with nodes and edges. In this work, we develop GMPSSG, a novel Gaussian Mixture-distributed Prototype mining framework for 3DSGG. Specifically, we model different categories with independent Gaussian Mixture-distributed Prototypes to effectively mitigate inter-class similarity, while employing multiple Gaussian components within each prototype to capture intra-class diversity. Moreover, Prototype-anchored Representation Learning is introduced to construct a well-structured and mutually independent category space; Topology-aware Prototype Interaction is devised to capture implicit co-occurrence priors within the scene, and leverage them to calibrate prototype distributions, thereby ensuring the plausibility of node-edge matching. Experiments on 3DSSG dataset demonstrate GMPSSG outperforms various top-leading methods. Our code is available at GMPSSG.

## 1. Introduction

The scene graph (SG) constructs abstract visual representations of scenes by modeling the relationships between objects and their surroundings. SG first emerged in the im-

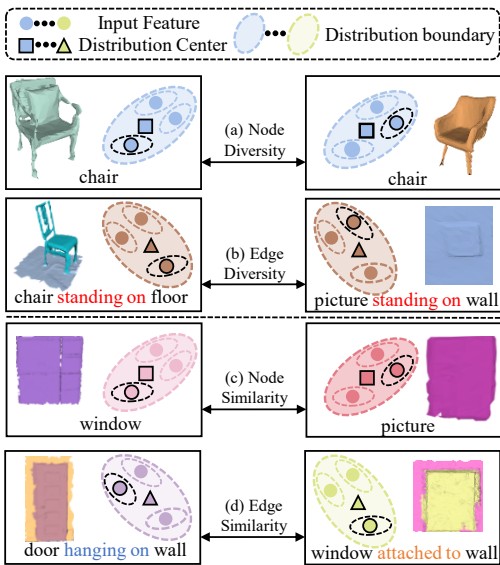

*Figure 1.* The illustration of nodes and edges with significant intra-class diversity and severe inter-class similarity.

age domain (Fu et al., 2025; Huang et al., 2025) and soon extends to video scene understanding (Chen et al., 2025; Li et al., 2025), continually driving richer and more comprehensive modeling of visual relationships. Recently, it has been introduced into 3D domain (Wu et al., 2023b; Zhang et al., 2024), serving as a key intermediate representation for various tasks such as robotic navigation (Werby et al., 2024; Yin et al., 2024), robot localization (Shaheer et al., 2023), and 3D scene reconstruction (Liu et al., 2025).

Existing 3D scene graph generation (3DSGG) methods typically rely on LiDAR point clouds as input (Lv et al., 2024; Heo et al., 2025). However, in practice, LiDAR sensors are resource-intensive and lack rich semantic details. Therefore, 3DSGG from RGB/RGB-D image sequences has gradually become a research hotspot (Feng et al., 2025b; Yeo et al., 2025). These methods typically reconstruct 3D point clouds from images and fuse multi-modal features (*e.g.*, images (Wu et al., 2023b; Yeo et al., 2025) and text (Feng et al., 2025a)) to extract and refine node and edge features for final prediction. Recent works have further introduced strong prior knowledge (*e.g.*, statistical prior (Yeo et al., 2025) and historical prior (Feng et al., 2025a)), to enhance prediction accuracy. Though impressive, these methods exhibit

---

[*]Equal contribution [1]School of Computer Science and Engineering, Nanjing University of Science and Technology, Nanjing, Jiangsu, China. Correspondence to: Xinguang Xiang <xgxiang@njust.edu.cn>, Xiangbo Shu <shuxb@njust.edu.cn>.

*Proceedings of the 43rd International Conference on Machine Learning*, Seoul, South Korea. PMLR 306, 2026. Copyright 2026 by the author(s).

two limitations: First, significant **intra-class diversity**. For nodes, objects within the same category (*e.g.*, chair) exhibit significant variations in structure and local details (Fig. 1a), leading to highly diverse visual appearances. For edges, the appearance of a relation often depends on the paired nodes (Fig. 1b). For example, *standing on* varies across triplets: "chair standing on floor" and "picture standing on wall" have very different spatial layouts and contact regions. However, many existing methods (Wu et al., 2023b; Feng et al., 2025a; Wu et al., 2021) model each category as one single prototype via discriminative predictors, which makes it hard to capture the distribution of node and edge features. Second, severe **inter-class similarity**. For nodes, objects from different categories often exhibit high overlap in the feature space (Fig. 1c). While windows and pictures possess different semantics, they share similar geometric structures, both appearing as planar rectangles attached to walls. For edges, the visual representations of different relationship categories are also extremely similar (Fig. 1d). For example, both "hanging on" and "attached to" involve objects placed on a vertical plane. However, existing methods (Feng et al., 2025b; Yeo et al., 2025) mainly focus on implicit feature enhancement and lack explicit discriminative mechanisms. This approach cannot effectively resolve the visual ambiguity caused by inter-class similarity.

To address these challenges, we propose GMPSSG, a Gaussian Mixture-distributed Prototype (GMP) mining framework for 3DSGG (Fig. 2). We model each node and edge category as an independent Gaussian Mixture-distributed Prototype to mitigate inter-class similarity, while employing multiple Gaussian components within each Gaussian Mixture-distributed Prototype to effectively characterize intra-class diversity. Specifically, different from previous approaches that focus on the adaptation of networks to robust node and edge feature extraction in complex 3D scenes, we accurately represent intra-class diversity and inter-class similarity by dynamically adjusting the component weights and corresponding means and variances of the prototypes.

In practice, following feature extraction, we model each category as a Gaussian Mixture-distributed Prototype. To enhance the discriminability of these prototypes, we introduce two learning mechanisms: (1) **Prototype-anchored Representation Learning,** which comprises three components designed to construct a well-structured and mutually independent category space. First, **Likelihood-driven Clustering Learning:** By maximizing the likelihood, this strategy guides node and edge features to converge towards their respective high-density regions, thereby enabling the prototypes to autonomously mine category-specific distributional characteristics. Second, **Mutual Information-guided Decorrelation Learning:** This mechanism introduces a distribution-level decorrelation mechanism. It effectively decouples visually similar node and edge features, thereby

mitigating the confusion induced by inter-class similarity. Third, **Cohesion-oriented Constraint Learning:** By imposing explicit constraints, this strategy compacts the prototype distributions of nodes and edges, further reducing interference from inter-class similarity. (2) **Topology-aware Prototype Interaction.** In 3D scenes, strong co-occurrence priors exist between specific nodes and edges. To capture these priors, we design explicit topological constraints to calibrate prototype distributions, thereby ensuring the plausibility of node-edge matching.

The contributions of this work can be summarized as follows: **First,** we revisit the 3D scene graph generation task from the perspective of feature distribution and design Gaussian Mixture-distributed Prototypes to explicitly characterize the intra-class diversity and inter-class similarity in nodes and edges, thereby enhancing the predictive capability of scene graphs. **Second,** we introduce Prototype-anchored Representation Learning. This strategy shapes prototypes from three dimensions, aiming to construct a well-structured and mutually independent category space, thereby achieving deep mining of node and edge feature distributions. **Third,** we design Topology-aware Prototype Interaction. By imposing explicit topological constraints, this strategy effectively captures implicit co-occurrence priors in the 3D scenes and calibrates prototype distributions via dynamic updating, thereby ensuring the plausibility of node-edge matching.

## 2. Related Work

**3D Scene Graph Generation (3DSGG).** 3DSGG aims to create a structured representation of 3D environment. Based on input modality, existing works are categorized into point cloud-based and RGB/RGB-D sequence-based methods.

Early point cloud-based methods can be categorized into two groups: the first focuses on contextual dependency modeling (Zhang et al., 2021a;b; Wang et al., 2023; Heo et al., 2025; Koch et al., 2024a; Ma et al., 2025), typically incorporating multi-dimensional edge features to explicitly model contextual dependencies; the second employs structure prior spatial reasoning (Feng et al., 2023a;b), leveraging the inherent hierarchy and layout constraints of indoor environments to regularize and refine relation predictions. Recently, driven by breakthroughs in vision-language models, the research focus has shifted towards open-vocabulary 3DSGG (Koch et al., 2024b; Chen et al., 2024; Zhang et al., 2025; Fei et al., 2026). These methods employ cross-modal alignment strategies to integrate textual semantics into geometric representations. However, in real-world applications, obtaining high-quality point clouds is limited by occlusions and sensor noise. Consequently, research has shifted toward methods based on RGB or RGB-D sequences.

Existing RGB/RGB-D sequence based methods adopt a

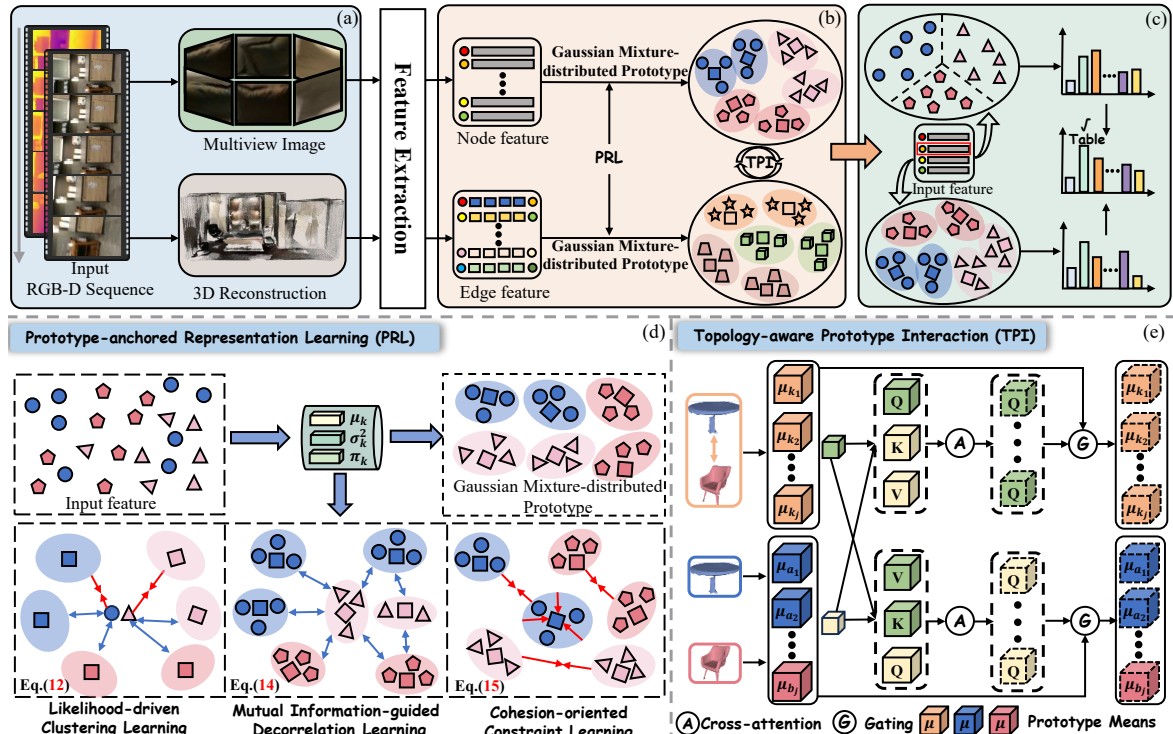

*Figure 2.* The overview of GMPSSG. (a) We first extract multi-view images and reconstruct 3D scenes from RGB-D image sequence. (b) To address intra-class diversity and inter-class similarity, after node and edge features extraction (§3.3), we describe each node and edge category with a set of Gaussian Mixture-distributed Prototypes (§3.4). (c) In the prediction phase, we incorporate the constructed prototypes and softmax probabilities to predict node and edge. (d) Prototype-anchored Representation Learning, which aims to construct a well-structured and independent prototype space (§3.5). (e) Topology-aware Prototype Interaction is devised to capture implicit co-occurrence priors within the scene, and calibrate prototype distributions, thereby ensuring the plausibility of node-edge matching (§3.6).

sequential pipeline that reconstructs point clouds prior to graph generation. Broadly, these methods can be summarized into two paradigms: the first focuses on representation enhancement (Wu et al., 2023b; Zhang et al., 2024; Wu et al., 2021; Feng et al., 2025b), integrating multi-modal information to extract more discriminative features; the second emphasizes inference refinement (Yeo et al., 2025; Feng et al., 2025a), incorporating strong prior knowledge to recalibrate and rescore predictions. Currently, emerging research explores (Hou et al., 2025) a new strategy that first parses 2D scene graphs from images and then lifts them into 3D space. Despite these advancements, existing methods largely overlook the key challenges posed by intra-class diversity and inter-class similarity in 3D scene graph generation.

**Gaussian Mixture Model.** Gaussian Mixture Model aims to represent data distributions through a mixture of Gaussian components. Recently, utilizing Gaussian Mixture Model to interpret inter-layer deep learning features has emerged as a prominent technique, which seeks to capture the intrinsic relationships between learned features and data distributions via probabilistic modeling. This approach results in more structured feature representations that explicitly capture class-conditional distributions, which could benefit subsequent tasks, *e.g.*, image segmentation (Wu et al.,

2023a; Liang et al., 2022), anomaly detection (Yao et al., 2024; Zavrtanik et al., 2021), object detection (Wang et al., 2025b; Luo et al., 2025), as well as provide a foundation for understanding model reasoning. Inspired by this, we design Gaussian Mixture-distributed prototypes for 3DSSG, which alleviates the intra-class diversity and inter-class similarity.

## 3. Methodology

### 3.1. Problem Definition

Formally, given an RGB-D image sequence, the objective of 3D scene graph generation can be defined as:

$$\mathcal{S} = (\mathcal{V}, \mathcal{E}), \tag{1}$$

where $\mathcal{V} = \{y_i\}_{i=1}^{M}$ denotes the set of $M$ object instances in the 3D scene, and $\mathcal{E} = \{z_{i \to j}\}$ denotes the set of directed relation edges defined on ordered object pairs $(i, j)$. Each node $y_i$ is associated with an object category label $y_i \in \{1, \ldots, c\}$, and each directed edge $(i \to j)$ is associated with a relation category label $z_{i \to j} \in \{1, \ldots, r\}$.

### 3.2. Overview

The overall framework of the network is shown in Fig. 2. We take a complete RGB-D image sequence as input and

follow the pipeline of ScenegraphFusion (Wu et al., 2021) to perform dense geometric reconstruction. We then apply 3D geometric segmentation (Tateno et al., 2017) to the reconstructed geometry, yielding a dense point cloud annotated with geometric segment labels. Meanwhile, we extract multi-view RGB images of the objects from the same sequence. We feed the reconstructed 3D point cloud with multi-view RGB images into feature encoding network to extract the final node and edge features (§3.3). These features are then used to build Gaussian Mixture-distributed Prototypes (GMP), which characterizes the class-conditional distributional properties of nodes and edges (§3.4). During training, we jointly update the Gaussian Mixture-distributed Prototype with two complementary learning strategies: (1) Prototype-anchored Representation Learning (§3.5); (2) Topology-aware Prototype Interaction (§3.6). During inference phase, following Wang et al. (2025b), we compute the posterior probability $p_1(c|\boldsymbol{x})$ of the Gaussian Mixture-distributed Prototype via Bayes' rule and fuse it with the predictive probability $p_2(c|\boldsymbol{x})$ from the discriminative branch. This allows us to infer the final node and edge categories.

### 3.3. Node and Edge Feature Initialization

Following MonoSSG (Wu et al., 2023b), we extract geometric features $\boldsymbol{g}_i$ via PointNet (Qi et al., 2017) and visual features $\boldsymbol{o}_i$ via DINOv2 (Oquab et al., 2023). Specifically, the initial node features are directly defined as the visual features $\boldsymbol{o}_i$, while the edge feature $\boldsymbol{e}_{i \to j}$ is derived from geometric attributes:

$$\boldsymbol{o}_i = \frac{1}{|\mathcal{K}_i|} \sum_{k \in \mathcal{K}_i} \mathtt{E}(\boldsymbol{I}_{i,k}), \qquad (2)$$

$$\boldsymbol{e}_{i \to j} = \mathtt{MLP}\Big( \big[ \boldsymbol{c}_j - \boldsymbol{c}_i, \ \boldsymbol{b}_j - \boldsymbol{b}_i, \ \boldsymbol{r}_{i \to j} \big] \Big), \qquad (3)$$

where, $\boldsymbol{I}_{i,k}$ is the image corresponding to node $i$ under the $k$-th observation, $\mathcal{K}_i$ denotes the set of available view indices for node $i$ and $\mathtt{E}(\cdot)$ denotes the image encoder; $\boldsymbol{c}_i, \boldsymbol{c}_j$ are the center-location of nodes, $\boldsymbol{b}_i, \boldsymbol{b}_j$ are their bounding-box parameter vectors, and $\boldsymbol{r}_{i \to j}$ is a relative pose descriptor which encodes the relative angle between two nodes.

Subsequently, we fuse the geometric features $\boldsymbol{g_i}$ into each node feature to obtain the new node feature $\hat{\boldsymbol{v}}_i$, then $\hat{\boldsymbol{v}}_i$ and $\boldsymbol{z}_{i \to j}$ are fed into multi-layer message passing, the Gated Recurrent Unit (GRU) is employed to update the node and edge features based on the messages. Specifically, the final node feature $\boldsymbol{v}_i$ and the final edge feature $\boldsymbol{z}_{i \to j}$ as follows:

$$\boldsymbol{v}_i^t = \mathtt{GRU}\big[ \boldsymbol{v}_i^{t-1}, \mathtt{MLP}(\hat{\boldsymbol{v}}_i, \max_{\mathbf{j} \in \mathcal{F}(\mathbf{i})} (\mathtt{FAN}(\hat{\boldsymbol{v}}_i, \boldsymbol{e}_{i \to j}, \hat{\boldsymbol{v}}_j))) \big], \quad (4)$$

$$\boldsymbol{z}_{i \to j}^t = \mathtt{GRU}\big[ \boldsymbol{z}_{i \to j}^{t-1}, \mathtt{MLP}((\hat{\boldsymbol{v}}_i, \ \boldsymbol{e}_{i \to j}, \ \hat{\boldsymbol{v}}_j)) \big], \qquad (5)$$

where, FAN denotes the feature-wise attention network (Wu et al., 2021), $\mathcal{F}(i)$ denotes the set of indices representing the neighboring nodes of $i$.

### 3.4. Gaussian Mixture-distributed Prototype

In 3D scenes, node and edge features often exhibit severe intra-class diversity and high inter-class similarity, which can lead to severe discriminative confusion. To address this issue, we design a Gaussian Mixture-distributed Prototype (GMP) mechanism to characterize the feature distributions. We model different categories with independent Gaussian Mixture-distributed Prototype to effectively mitigate inter-class similarity, while employing multiple Gaussian components within each prototype to capture intra-class diversity.

Specifically, for a set of node or edge features $\boldsymbol{X} = \{\boldsymbol{x}_1, \ldots, \boldsymbol{x}_n\}$, we learn a set of Gaussian Mixture-distributed Prototypes $\{\mathcal{G}_c\}_{c=1}^C$ to model the feature distribution of each class. The prototypes employ a weighted mixture of $m$ Gaussian components for modeling the distribution of each class $c$ in the $d$-dimensional embedding space:

$$p(\boldsymbol{x}_i \mid c) = \sum_{k=1}^K \boldsymbol{\pi}_{c,k} \mathcal{N}(\boldsymbol{x}_i, \boldsymbol{\mu}_{c,k}, \boldsymbol{\sigma}_{c,k}), \qquad (6)$$

where, $p(\boldsymbol{x}_i \mid c)$ denotes the class-conditional probability density of observing the sample $x_i$ given class $c$. $\boldsymbol{\pi}_{c,k}$ is the mixture weight of the $m$-th Gaussian component, which measures its relative contribution to the mixture distribution of class $c$, satisfying $\boldsymbol{\pi}_{c,k} \geq 0$ and $\sum_{k=1}^K \boldsymbol{\pi}_{c,k} = 1$. $\mathcal{N}(\cdot)$ denotes the probability density value assigned to $x_i$ by the $m$-th Gaussian component, which can be computed as:

$$\mathcal{N}(\boldsymbol{x_i}; \boldsymbol{\mu}_{c,k}, \boldsymbol{\sigma}_{c,k}) =$$
$$\frac{1}{\sqrt{(2\pi)^N |\boldsymbol{\sigma}_{c,k}|}} \exp\left( -\frac{1}{2} (\boldsymbol{x}_i - \boldsymbol{\mu}_{c,k})^\top \boldsymbol{\sigma}_{c,k}^{-1} (\boldsymbol{x}_i - \boldsymbol{\mu}_{c,k}) \right), \tag{7}$$

where $\boldsymbol{\mu}_{c,k}$ denotes the mean of the $k$-th Gaussian component for class $c$, $\boldsymbol{\sigma}_{c,k}$ is the corresponding covariance matrix.

Based on the class-conditional probability density $p(\boldsymbol{x}_i \mid c)$ along with the class prior $p(c)$, we can apply Bayes' theorem to compute the posterior probability as follows:

$$p(c \mid \boldsymbol{x}_i) = \frac{p(\boldsymbol{x}_i \mid c) \, p(c)}{\sum_{c'=1}^C p(\boldsymbol{x}_i \mid c') \, p(c')} = \frac{p(\boldsymbol{x}_i \mid c)}{\sum_{c'=1}^C p(\boldsymbol{x}_i \mid c')}, \tag{8}$$

by assuming $p(c) = 1/c$, as commonly adopted in the literature (Wang et al., 2025a; Liang et al., 2022).

### 3.5. Prototype-anchored Representation Learning

Gaussian Mixture-distributed Prototype (GMP) is parameterized by three sets of parameters: $\boldsymbol{\phi}_c = (\boldsymbol{\mu}_{c,k}, \boldsymbol{\sigma}_{c,k}, \boldsymbol{\pi}_{c,k})$. A straightforward approach to estimating the parameters $\boldsymbol{\phi}_c$ is to employ the standard Expectation–Maximization (EM) algorithm (Liang et al., 2022; Zhu et al., 2025; Song et al., 2024; Xu et al., 2024; Ronen et al., 2022).

In the E-step, we compute the responsibilities for each node and edge feature as follows:

$$\gamma_{c,k} = \frac{\pi_{c,k}^{(t-1)} \mathcal{N}\left(\boldsymbol{x}_i, \boldsymbol{\mu}_{c,k}^{(t-1)}, \boldsymbol{\sigma}_{c,k}^{(t-1)}\right)}{\sum_{k'=1}^{K} \pi_{c,k'}^{(t-1)} \mathcal{N}\left(\boldsymbol{x}_i, \boldsymbol{\mu}_{c,k'}^{(t-1)}, \boldsymbol{\sigma}_{c,k'}^{(t-1)}\right)}, \qquad (9)$$

where, $\gamma_{c,k}$ denotes the posterior responsibility. The standard EM algorithm provides a closed-form M-step solution to estimate the parameters:

$$\begin{cases} \boldsymbol{\pi}_{c,k}^{(t)} = \dfrac{N_{c,k}^{(t)}}{N_c}, \boldsymbol{\mu}_{c,k}^{(t)} = \dfrac{1}{N_{c,k}^{(t)}} \sum_{\boldsymbol{x}_i = c} \gamma_{c,k}\, \boldsymbol{x}_i, \\[2mm] \boldsymbol{\sigma}_{c,k}^{(t)} = \dfrac{1}{N_{c,k}^{(t)}} \sum_{\boldsymbol{x}_i = c} \gamma_{c,k}\big(\boldsymbol{x}_i - \boldsymbol{\mu}_{c,k}^{(t)}\big)\big(\boldsymbol{x}_i - \boldsymbol{\mu}_{c,k}^{(t)}\big)^{\top}, \end{cases} \qquad (10)$$

where $N_{c,k}$ denotes the number of training samples labeled as $c$, and $N_{c,k}^{(t)} = \sum_{n:\, c_n = c} \gamma_{c,k}$.

While efficient, the standard EM algorithm's closed-form M-step updates induce intra-class representation collapse and inter-class spatial overlap. This limitation is particularly pronounced in 3D scenes characterized by severe intra-class diversity and inter-class similarity. To address this, we introduce prototype-anchored representation learning to reshape parameter updates by leveraging feature space structure, incorporating three core mechanisms: likelihood-driven clustering learning, mutual information-guided decorrelation learning, and cohesion-oriented constraint learning.

**Likelihood-driven Clustering Learning.** We expect node and edge features within the same class to be well characterized by a single class-specific Gaussian Mixture-distributed Prototype. To achieve this, we revisit the EM algorithm: from an optimization perspective, EM essentially maximizes the class-conditional log-likelihood over the class-specific sample set $\boldsymbol{X}_c = \{x_i \in \boldsymbol{X} \mid x_i = c\}$, encouraging samples belonging to class $c$ to attain higher probability under their class-conditional distribution. This can be formulated as:

$$\phi_c^* = \arg\max_{\phi_c} \log\left[\prod_{i=1}^{m} \sum_{k=1}^{K} \boldsymbol{\pi}_{c,k} \mathcal{N}(\boldsymbol{x}_i, \boldsymbol{\mu}_{c,k}, \boldsymbol{\sigma}_{c,k})\right], \quad (11)$$

Building on this, we adopt the class-conditional negative log-likelihood of the training samples as the optimization objective, using it to drive features of the same class $c$ to be modeled more stably and consistently by the corresponding Gaussian Mixture-distributed prototype. Specifically, we optimize the parameters $\phi_c$ by minimizing the following class-conditional negative log-likelihood loss:

$$\mathcal{L}_{\text{LCL}} = -\frac{1}{N} \sum_{c=1}^{C} \sum_{i=1}^{N_c} \mathbb{I}_{i:c} \log\big(p(\boldsymbol{x}_i \mid c)\big), \qquad (12)$$

where $N$ and $C$ denote the total number of features and the total number of classes in the batch, respectively. $N_c$ denotes the number of features of class $c$ in the batch. $\mathbb{I}_{i:c}$ is a binary indicator function that matches the class label of the current iteration with the $c$-th class label; it is set to 1 if the input sample belongs to class $c$ and 0 otherwise.

**Mutual Information-guided Decorrelation Learning.**

Although likelihood-driven clustering learning fits distributions well, the lack of explicit separation often causes prototype overlap, especially in 3D scenes with high inter-class similarity. From an information-theoretic perspective, enhancing discriminability is equivalent to maximizing the mutual information $\mathrm{I}(c; x)$ (Alemi et al., 2016), which corresponds to stronger distributional separability (Boudiaf et al., 2020). Therefore, we design a distributional separation regularizer to enlarge the discrepancy among prototypes.

To quantify such discrepancy, we employ the Kullback–Leibler (KL) divergence to measure the distributional discrepancy across different classes. However, we do not directly compute the symmetric KL divergence between Gaussian Mixture-distributed prototypes, since the KL divergence between two Gaussian Mixture-distributed prototypes has no simple closed-form solution and may introduce additional computational cost. Instead, for each class, we first fuse its Gaussian components into a class-level diagonal Gaussian distribution via moment matching. In practice, we adopt the symmetric KL divergence, which is defined as:

$$\mathrm{D}(p_c, p_{c'}) = \frac{1}{2}\Big(\mathbb{D}_{\text{KL}}\big(p(\boldsymbol{x} \mid c) \,\|\, p(\boldsymbol{x} \mid c')\big) + \mathbb{D}_{\text{KL}}\big(p(\boldsymbol{x} \mid c') \,\|\, p(\boldsymbol{x} \mid c)\big)\Big), \qquad (13)$$

where $p_c$ and $p_{c'}$ denote the Gaussian mixture distribution corresponding to class $c$ and class $c'$, respectively. Based on this metric, we impose inter-class repulsion by directly acting on the Gaussian mixture-distributed prototypes of each class, which can be defined as follows:

$$\mathcal{L}_{\text{MDL}} = -\log \frac{1}{\sum_{c^- \in \mathcal{O}(c)} \exp(-\mathrm{D}(p_c, p_{c^-})/\tau) + 1}, \quad (14)$$

where $c^-$ denotes the class that is irrelevant to class $c$. $\mathcal{O}(c)$ denotes the set of negative classes, and $\tau > 0$ is a temperature hyper-parameter. Notably, for nodes, we construct the negative set by including all other nodes as well as all edges, thereby ensuring sufficient scale and diversity of negative samples; for edges, we adopt the same strategy.

**Cohesion-oriented Constraint Learning.** By likelihood-driven clustering learning and mutual information-guided decorrelation learning, we construct the Gaussian Mixture-distributed prototypes and explicitly enlarge the discrepancy among class-specific prototypes. However, without proper constraints, multiple Gaussian components of the same class may lead to intra-class representations that are discontinuous and not compact enough. Therefore, we further impose structured regularization on the Gaussian components within each prototype, constraining their geometric properties from both the $\mu$ and $\sigma$ to encourage intra-class continuity and compactness, which can be defined as follows:

$$\mathcal{L}_{\text{CCL}} = \sum_{c=1}^{C} \sum_{m=1}^{K} \sum_{n=m+1}^{K} \left( \sqrt{\sum_{\ell=1}^{d} \left( \boldsymbol{\mu}_{c,m}^{(\ell)} - \boldsymbol{\mu}_{c,n}^{(\ell)} \right)^2} + \boldsymbol{\sigma}_{c,m} \right),$$
$$\tag{15}$$

where $\boldsymbol{\mu}_{c,m}$ and $\boldsymbol{\mu}_{c,n}$ respectively denote the mean vectors of two different Gaussian components within the same class $c$, and $\boldsymbol{\sigma}_{c,m}$ denotes the variance parameter associated with the $m$-th Gaussian component in class $c$.

**Overall Loss Function.** The overall training loss function is the combination of the Eq. 12, Eq. 14, Eq. 15, as follows:

$$\mathcal{L} = \lambda_1 \mathcal{L}_{node} + \lambda_2 \mathcal{L}_{edge} + \lambda_3 \mathcal{L}_{\text{LCL}} + \lambda_4 (\mathcal{L}_{\text{MDL}} + \mathcal{L}_{\text{CCL}}), \tag{16}$$

where $\mathcal{L}_{node}$ and $\mathcal{L}_{edge}$ denote the cross-entropy loss for node and edge classification. $\lambda_1$ and $\lambda_2$ are dynamically computed based on the number of nodes and edges in the current scene, $\lambda_3 = 0.1$, $\lambda_4 = 0.05$.

### 3.6. Topology-aware Prototype Interaction

3D scene graphs exhibit a strong co-occurrence prior: the co-occurrence of subject-object pairs and their relations is governed by high determinacy. For example, the subject–object pair *table–chair* is rarely compatible with relations such as *part of* or *standing on* under typical scene statistics; conversely, the relation *standing on* induces a strong topological bias that makes the pair *chair–window* highly unlikely.

To leverage the co-occurrence priors in 3D scene graphs, we instantiate topological constraints as supervisory signals, which in turn impose manifold regularization within the mean space of the prototypes. By anchoring distributional centroids to a feasible space aligned with the priors during iteration, this mechanism suppresses topological inconsistencies at their geometric origin. Specifically, we first model the co-occurrence priors of node class $c$ and edge class $r$ as latent knowledge offsets, $\boldsymbol{\Delta}_c^n$ and $\boldsymbol{\Delta}_r^e$, respectively. Their explicit forms are formulated as follows:

$$\boldsymbol{\Delta}_c^n = \text{MHA}(\bar{\boldsymbol{\mu}}_c^n, \bar{\boldsymbol{\mu}}_r^e, \bar{\boldsymbol{\mu}}_r^e), \boldsymbol{\Delta}_r^e = \text{MHA}(\bar{\boldsymbol{\mu}}_r^e, \bar{\boldsymbol{\mu}}_c^n, \bar{\boldsymbol{\mu}}_c^n), \tag{17}$$

where $\bar{\boldsymbol{\mu}}_c^e$ is the key set for node $c$, aggregated from its associated edge prototypes. Conversely, $\bar{\boldsymbol{\mu}}_r^n$ is the key set for relation $r$, aggregated from its two endpoint node prototypes. Their explicit forms are given as follows:

$$\bar{\boldsymbol{\mu}}_c^n = \left\{ \text{w}_{n \to e} [\bar{\boldsymbol{\mu}}_{s_t}^n ; \bar{\boldsymbol{\mu}}_{o_t}^n] \right\}_{t=1}^{T_r}, \bar{\boldsymbol{\mu}}_r^e = \left\{ \text{w}_{e \to n} [\bar{\boldsymbol{\mu}}_{r_t}^e] \right\}_{t=1}^{T_c}, \tag{18}$$

where, $\text{w}(\cdot)$ denotes an aggregation operator, while $s_t, o_t \in \mathcal{V}$ and $r_t \in \mathcal{E}$ represent the sets of subject nodes, object nodes, and their corresponding relations observed in the current scene. During optimization, we introduce an online gating mechanism that adaptively controls the injection of knowledge. Specifically, the updated mean vectors of the node and edge GMP, $\tilde{\boldsymbol{\mu}}_{c,m}^n$ and $\tilde{\boldsymbol{\mu}}_{r,m}^e$, are given by:

$$\begin{cases} \tilde{\boldsymbol{\mu}}_{c,m}^n \leftarrow \text{G}_c^n \odot \boldsymbol{\mu}_c^n + (1 - \text{G}_c^n) \odot \boldsymbol{\Delta}_c^n, \\ \tilde{\boldsymbol{\mu}}_{r,m}^e \leftarrow \text{G}_r^e \odot \boldsymbol{\mu}_r^e + (1 - \text{G}_r^e) \odot \boldsymbol{\Delta}_r^e, \end{cases} \tag{19}$$

where, $\text{G}_c^n, \text{G}_r^e \in (0, 1)$ denote the gating coefficients. Notably, we employ only the mean vector $\boldsymbol{\mu}$ of the Gaussian Mixture-distributed prototype to model the prior. Since $\boldsymbol{\mu}$ represents the core position of the Gaussian component in the feature space, it encodes the structural consistency between nodes and edges more stably than other parameters.

## 4. Experiments

### 4.1. Experimental Setup

**Dataset.** We conduct experiments on the widely used 3DSGG benchmark 3DSSG (Wald et al., 2020). Built upon the 3RScan dataset (Wald et al., 2019), 3DSSG provides annotated 3D semantic scene graphs. We report results using both reconstructed dense point clouds (Dense) and ground-truth point clouds (GT) as inputs. For the Dense setting, following the ScanNet benchmark, we map the labels to a subset of 20 node classes and 8 edge classes. For the GT setting, we evaluate our model with this same mapping and further report performance on the full label set, comprising 160 node classes and 26 edge classes.

**Evaluation Metric.** We evaluate our model across three standard task dimensions: (1) Triplet Classification, (2) Object Classification, (3) Predicate Classification. Following the standard evaluation protocol in the field (Wu et al., 2023b), we adopt Top-1 Recall (R@1) and Mean Recall@1 (mR@1) as our primary metrics.

**Implementation Details.** All experiments are conducted on a single NVIDIA GeForce RTX 4090 GPU with a batch size of 1, using an early-stopping criterion implemented in PyTorch. GMPSSG is trained end-to-end with AdamW optimizer, starting from a learning rate of $1 \times 10^{-4}$. More details are provided in §A.3 of Appendix.

### 4.2. Comparison with State-of-the-Art Methods

**Performance on Dense Point Cloud Setting.** As summarized in the upper part of Table 1, our method outperforms state-of-the-art (SOTA) methods under the dense point cloud setting. Concretely, our method yields significant gains on both Triplet and Object metrics, boosting Tri. to **43.0** (**+2.5%**) and Obj. to **64.8** (**+3.0%**). Notably, our method also achieves improvements after excluding "none" relationship, reaching **30.6** (**+4.9%**) on Tri.* and **66.4** (**+5.3%**) on Obj.*. Besides, regarding predicate prediction, our method achieves the highest score on Pred.*, reaching **33.2** (**+5.6%**). These improvements are attributed to the Gaussian Mixture-distributed prototypes, which not only captures intra-class diversity but also mitigates inter-class similarity.

**Performance on GT Point Cloud Setting.** Tables 1 and 2 report quantitative comparisons using ground-truth point clouds as input. In the lower part of Table 1, our method

*Table 1.* **Quantitative results** (§4.2) on 3DSSG (Wald et al., 2020) within 20 node and 8 edge classes, which from reconstructed dense and ground truth point clouds. The best and the second best results are highlighted in bold and underlined. ∗: without *none* relationship.

| Method | Input | Recall(%) | | | | | | mRecall(%) | |
|---|---|---|---|---|---|---|---|---|---|
| | | Tri.↑ | Tri.* ↑ | Obj.↑ | Obj.* ↑ | Pred.↑ | Pred.* ↑ | Obj.↑ | Pred.↑ |
| IMP (Xu et al., 2017)[CVPR2017] | Dense | 25.4 | 14.4 | 47.7 | 48.1 | 89.9 | 16.8 | 35.8 | 19.9 |
| VGfM (Gay et al., 2018)[ACCV2018] | Dense | 27.4 | 17.0 | 50.1 | 51.3 | **90.6** | 19.8 | 33.6 | 22.3 |
| SGPN (Wald et al., 2020)[CVPR2020] | Dense | 13.0 | 11.9 | 35.6 | 36.2 | 86.8 | 25.7 | 26.0 | 24.3 |
| SGFN (Wu et al., 2021)[CVPR2021] | Dense | 29.9 | 21.5 | 52.2 | 53.8 | 89.5 | 24.6 | 35.9 | 27.1 |
| MonoSSG (Wu et al., 2023b)[CVPR2023] | Dense | 32.9 | 23.1 | 54.5 | 55.1 | 88.7 | 26.6 | 45.4 | 35.2 |
| HEDSGP (Feng et al., 2025b)[TPAMI2025] | Dense | 31.3 | - | 55.5 | - | 90.1 | - | 47.3 | 36.9 |
| SCRSSG (Yeo et al., 2025)[ICCV2025] | Dense | 40.5 | 25.7 | 61.8 | 61.1 | 90.4 | 27.6 | **60.5** | 39.2 |
| GMPSSG (Ours) | Dense | **43.0** | **30.6** | **64.8** | **66.4** | 88.3 | **33.2** | 59.3 | **42.0** |
| IMP (Xu et al., 2017)[CVPR2017] | GT | 45.3 | 44.3 | 65.4 | 65.9 | 94.0 | 56.6 | 56.2 | 41.8 |
| VGfM (Gay et al., 2018)[ACCV2018] | GT | 52.9 | 51.4 | 70.7 | 71.3 | 94.9 | 62.8 | 59.5 | 46.8 |
| SGPN (Wald et al., 2020)[CVPR2020] | GT | 31.8 | 39.7 | 55.1 | 55.6 | 95.4 | 71.0 | 47.7 | 61.5 |
| SGFN (Wu et al., 2021)[CVPR2021] | GT | 42.6 | 47.6 | 63.6 | 64.4 | 94.2 | 68.9 | 53.6 | 63.1 |
| VL-SAT (Wang et al., 2023)[CVPR2023] | GT | 66.1 | - | 78.4 | - | 89.9 | - | 53.8 | 67.1 |
| MonoSSG (Wu et al., 2023b)[CVPR2023] | GT | 63.8 | 63.4 | 79.3 | 80.0 | 95.6 | 76.0 | 78.2 | 64.8 |
| EgoSG (Zhang et al., 2024)[CVPRW2024] | GT | 61.1 | - | 56.4 | - | 78.6 | - | 48.6 | 66.7 |
| HESGR (Feng et al., 2025a)[TCVST2025] | GT | 67.6 | - | 83.3 | - | **96.5** | - | 69.1 | 71.2 |
| HEDSGP (Feng et al., 2025b)[TPAMI2025] | GT | 66.1 | - | 83.2 | - | 96.1 | - | 78.9 | 68.8 |
| GMPSSG (Ours) | GT | **73.0** | **69.9** | **84.5** | **85.2** | 95.8 | **79.4** | **85.9** | **73.4** |

*Table 2.* **Quantitative results** (§4.2) on 3DSSG (Wald et al., 2020) within 160 node and 26 edge classes, which from ground-truth point cloud. The best and the second best results are highlighted in bold and underlined.

| Method | Recall(%) | | | mRecall(%) | |
|---|---|---|---|---|---|
| | Tri.↑ | Obj.↑ | Pred.↑ | Obj.↑ | Pred.↑ |
| IMP (Xu et al., 2017) | 64.1 | 42.9 | 16.2 | 16.0 | 3.6 |
| VGfM (Gay et al., 2018) | 64.5 | 45.9 | 17.4 | 19.1 | 5.5 |
| SGPN (Wald et al., 2020) | 64.8 | 27.9 | **67.0** | 12.1 | 20.9 |
| SGFN (Wu et al., 2021) | 64.7 | 36.9 | 48.4 | 16.2 | 14.4 |
| MonoSSG (Wu et al., 2023b) | 67.6 | 53.4 | 48.1 | 28.9 | 24.7 |
| SCRSSG (Yeo et al., 2025) | 57.7 | 32.1 | 10.9 | 19.6 | 2.3 |
| GMPSSG (Ours) | **68.6** | **59.9** | 49.1 | **37.3** | **29.6** |

*Table 3.* **Analysis of core components** (§4.3) on 3DSSG (Wald et al., 2020) with dense point cloud setting.

| Base -line | GMP | PRL | TPI | Recall(%) | | | mRecall(%) | |
|---|---|---|---|---|---|---|---|---|
| | | | | Tri.↑ | Obj.↑ | Pred.↑ | Obj.↑ | Pred.↑ |
| ✓ | | | | 35.0 | 59.6 | 86.9 | 55.9 | 38.6 |
| ✓ | ✓ | ✓ | | 42.2 | 63.9 | **88.8** | 59.0 | 41.0 |
| ✓ | ✓ | ✓ | ✓ | **43.0** | **64.8** | 88.3 | **59.3** | **42.0** |

achieves the best performance on Tri. and Tri.*, reaching **73.0** (+**5.4**%) and **69.9** (+**6.5**%), respectively. Under the larger-scale setting in Table 2, where the substantially increased numbers of node and edge categories make 3D scene graph generation more challenging, our approach also yields strong gains in mean object recall and mean predicate recall, achieving **37.3** (+**8.4**%) and **29.6** (+**4.9**%), respectively. These results validate the generalization of our method.

### 4.3. Ablation Study

To evaluate our algorithmic designs and gain deeper insights, we conduct ablation studies on 3DSSG (Wald et al., 2020) under the reconstructed dense point cloud setting with 20 node classes and 8 edge classes.

**Key Component Analysis.** We first validate the effectiveness of our core concept, Gaussian Mixture-distributed Prototypes (GMP), alongside the proposed training strategies: Prototype-anchored Representation Learning (PRL, §3.5) and Topology-aware Prototype Interaction (TPI, §3.6). As shown in Table 3, the baseline performs discriminative classification directly on extracted features. Integrating GMP with PRL yields substantial improvements, notably boosting Tri. by **7.2**%. The subsequent introduction of TPI brings further gains. These results demonstrate that PRL and TPI effectively enable GMP to capture the distributional structure of the feature space, thereby better modeling intra-class diversity and alleviating inter-class similarity.

**Key Component of PRL (§3.5).** Next, we analyze three key training losses in Prototype-anchored Representation Learning. As shown in Table 4, introducing $\mathcal{L}_{\text{LCL}}$ yields a clear improvement, boosting Tri. by **6.2**% over the baseline. Building on this, further adding $\mathcal{L}_{\text{MDL}}$ (Eq. 14) and $\mathcal{L}_{\text{CCL}}$ (Eq. 15) leads to additional gains, resulting in a final Tri. score of **42.2**. Overall, these results confirm that the proposed objectives are complementary and jointly help learn more structured and discriminative prototype distributions.

**Numbers of Gaussian Components.** We study the impact of the number of Gaussian components per GMP, with results reported in Table 5. Overall, increasing the number of components yields consistent performance gains, supporting our hypothesis that multiple components help capture diverse intra-class patterns for both nodes and edges. However, when the component number exceeds $K = 4$, performance starts to drop (*i.e.*, Tri: **43.0**→39.0). We suspect that this is due to over-parameterization, which introduces redundant components and unstable estimation, thereby weakening the effectiveness of prototype modeling.

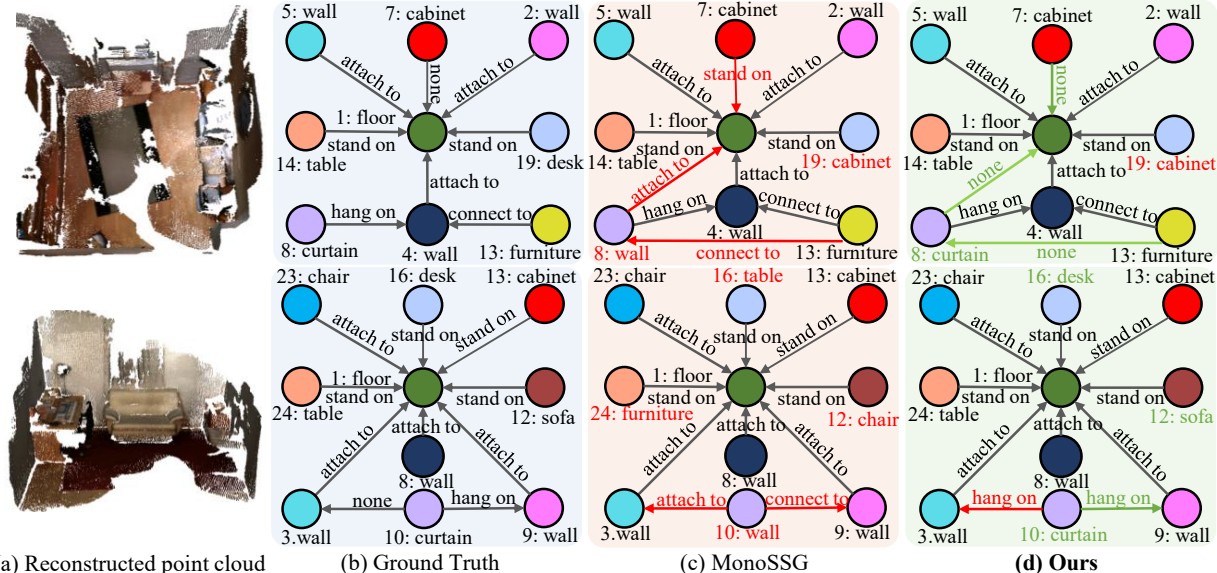

| (a) Reconstructed point cloud | (b) Ground Truth | (c) MonoSSG | (d) Ours |

*Figure 3.* Qualitative results from MonoSSG (Wu et al., 2023b) and our method on the 3DSSG (Wald et al., 2020) dataset. Red edge: misclassified edges from MonoSSG, green edge: edges corrected by our method, red node: misclassified node.

*Table 4.* **Analysis of representation learning components** (§4.3) on 3DSSG (Wald et al., 2020) with dense point cloud setting.

| GMP | $\mathcal{L}_{LCL}$ Eq.12 | $\mathcal{L}_{CCL}$ Eq.15 | $\mathcal{L}_{MDL}$ Eq.14 | Recall(%) Tri.↑ | Obj.↑ | Pred.↑ | mRecall(%) Obj.↑ | Pred.↑ |
|---|---|---|---|---|---|---|---|---|
| ✓ | ✓ | | | 41.2 | 63.1 | 88.8 | 58.4 | 40.6 |
| ✓ | ✓ | ✓ | | 41.6 | 63.4 | **89.6** | 58.5 | 40.4 |
| ✓ | ✓ | ✓ | ✓ | **42.2** | **63.9** | 88.8 | **59.0** | **41.0** |

*Table 5.* **Analysis of different numbers of Gaussian components** (§4.3) on 3DSSG (Wald et al., 2020) with dense point cloud setting.

| Number of Components | Recall(%) Tri.↑ | Obj.↑ | Pred.↑ | mRecall(%) Obj.↑ | Pred.↑ |
|---|---|---|---|---|---|
| 0 | 35.0 | 59.6 | 86.9 | 55.9 | 38.6 |
| 1 | 39.6 | 63.2 | 85.2 | 59.9 | 40.8 |
| 2 | 40.5 | 64.2 | 84.4 | **60.1** | 41.0 |
| 3 | 42.5 | 64.1 | 88.0 | 59.4 | 40.3 |
| 4 | **43.0** | **64.8** | **88.3** | 59.3 | **42.0** |
| 5 | 39.0 | 63.9 | 84.8 | 59.5 | 41.4 |

*Table 6.* **Ablation study of different update strategies** (§4.3) on 3DSSG (Wald et al., 2020) with dense point cloud setting.

| Update Strategy | Recall(%) Tri.↑ | Obj.↑ | Pred.↑ | mRecall(%) Obj.↑ | Pred.↑ |
|---|---|---|---|---|---|
| Standard EM | 41.1 | 62.9 | 87.8 | 58.2 | 40.6 |
| Sinkhorn EM | 41.4 | 63.1 | 86.9 | 58.5 | 40.2 |
| Our | **43.0** | **64.8** | **88.3** | **59.3** | **42.0** |

**Update Strategy.** We evaluate the effectiveness of our proposed update strategy by comparing it with other methods: the standard EM algorithm (Kirilenko et al., 2024) and the Sinkhorn-EM algorithm (Liang et al., 2022). As shown in Table 6, our update strategy achieves the best performance and consistently outperforms both Standard EM and Sinkhorn-EM across all metrics, *e.g.*, **+1.9%**, **+1.6%** on Tri., which shows that incorporating prior guidance facilitates parameter optimization, yielding better prediction.

**Interaction Strategy.** We further evaluate the effect of different interaction strategies, including node-to-edge uni-

*Table 7.* **Ablation study of different interaction strategies** (§4.3) on 3DSSG (Wald et al., 2020) with dense point cloud setting.

| Base -line | $p_{n \to e}$ | $p_{e \to n}$ | Recall(%) Tri.↑ | Obj.↑ | Pred.↑ | mRecall(%) Obj.↑ | Pred.↑ |
|---|---|---|---|---|---|---|---|
| ✓ | | | 42.2 | 63.9 | 88.8 | 59.0 | 41.0 |
| ✓ | ✓ | | 42.4 | 64.2 | **89.2** | 59.2 | 40.6 |
| ✓ | | ✓ | 42.5 | 64.2 | 88.5 | **59.7** | 40.8 |
| ✓ | ✓ | ✓ | **43.0** | **64.8** | 88.3 | 59.3 | **42.0** |

directional interaction $p_{n \to e}$, edge-to-node unidirectional interaction $p_{e \to n}$, and bidirectional interaction between nodes and edges. As shown in Table 7, either unidirectional interaction yields consistent improvements, while bidirectional interaction brings additional gains. These results suggest that modeling node–edge interactions effectively leverages structural priors in scene graphs, leading to better prediction.

### 4.4. Qualitative Analysis

We present the qualitative comparison between MonoSSG and our method in Fig. 3. The results demonstrate that our method generates more semantically reliable scene graphs, outperforming the MonoSSG in the accuracy of both nodes and edges. Specifically, our method effectively overcomes visual ambiguity, accurately discriminating between nodes with similar textures (*e.g.*, curtain and wall). Meanwhile, it excels in fine-grained semantic relationship analysis, successfully disambiguating confused predicates (*e.g.*, connected to and hanging on). Interestingly, our method significantly suppresses the over-prediction of invalid relationships. More visualizations are provided in §C of Appendix.

### 5. Conclusion

In this work, we propose GMPSSG, a Gaussian Mixture-distributed Prototype mining framework for 3D Scene Graph

Generation, which aims to mitigate the challenges of intra-class diversity and inter-class similarity by constructing prototypes. Specifically, it employs the following two strategies for updating: (1) Prototype-anchored Representation Learning, designed to construct a well-structured and independent category space; (2) Topology-aware Prototype Interaction is devised to capture co-occurrence priors within the scene, thereby ensuring the plausibility of node-edge matching.

## Acknowledgments

This work is supported by the National Natural Science Foundation of China (No. U25A20442, 62272230, and 62427808).

## Impact Statement

This paper presents work whose goal is to advance the field of Machine Learning. There are many potential societal consequences of our work, none which we feel must be specifically highlighted here.

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

- §A Detailed Experimental Setup.

- §B More Quantitative Results.

- §C More Qualitative Visualization.

- §D More Discussion.

## A. Detailed Experimental Setup

### A.1. Dataset Description

Built upon the real-world 3RScan benchmark (Wald et al., 2019), the 3DSSG dataset (Wald et al., 2020) comprises 1335 reconstructed indoor scenes represented as 3D point clouds with instance-level annotations, associated with approximately 363k RGB-D frames. Notably, this dataset poses significant challenges: as the images were acquired using consumer-level devices, the sequences typically suffer from low frame rates (10 Hz), motion blur, and image jitter caused by sudden camera movements. Furthermore, each scene is essentially composed of multiple short video clips with varying motion trajectories spliced together; this discontinuity and the low quality of the visual data further exacerbate the difficulty of the task. In terms of scale, the dataset encompasses 160 node categories and 26 edge categories, containing a total of 48K nodes and 544K edges. Following the standard split, we partition the dataset into 1061 training scans, 117 validation scans, and 157 test scans.

### A.2. Evaluation

**Evaluation Tasks.** (1) **Object Classification:** This task aims to assign the correct semantic label to each node within the scene graph, such as identifying a "chair" or "table". It evaluates the accuracy of entity recognition alone, independent of the prediction results for the connections between nodes. (2) **Predicate Classification:** This task focuses on predicting the semantic relationships embedded within the edges connecting node pairs, such as distinguishing between "standing on" and "supported by". It primarily evaluates the model's capability to reason about object interactions, independent of the classification accuracy of the object labels themselves. (3) **Triplet Classification:** This task involves the joint prediction of the triplet and serves as the most rigorous metric for holistic scene understanding. Under this criterion, a prediction is deemed correct only if the subject class, object class, and the connecting relationship predicate all fully match the ground truth.

**Evaluation Metrics.** For a fair comparison, we follow the protocol established by Wu et al. (Wu et al., 2023b), reporting results both with and without the 'none' category for a comprehensive evaluation. Notably, our *recall@1* metric differs from the standard *recall@N* typically used in 2D scene graph tasks (Fu et al., 2025; Huang et al., 2025; Chen et al., 2025; Li et al., 2025). While the standard approach selects the top $N$ triplets from a global ranking, our metric evaluates the accuracy of the single most confident prediction (subject, predicate, object) for each detected triplet. Given that our model produces an unconstrained number of triplets, this metric serves as a practical proxy for *recall@$\infty$* under graph constraints. For simplicity, we refer to this metric as *recall* throughout the paper.

### A.3. Training Details

For each Gaussian Mixture-distributed Prototype, we initialize its mean, variance, and mixture weight separately. The means are initialized by a truncated normal distribution with $\sigma = 0.02$, which introduces small differences among Gaussian components and prevents identical initial states. The variance parameters are initialized to 0 and transformed by a softplus function with a lower bound $\sigma_{\min} = 0.03$, ensuring positive and stable variances. The mixture weights are parameterized by learnable logits initialized to 0, and the softmax function gives uniform initial weights for all components. In terms of network configuration, we set PointNet to be trainable, while freezing the parameters of DINOv2 to directly utilize its pre-trained weights without fine-tuning. For multi-view image inputs, all images are resized to a resolution of $224 \times 224$ and subsequently fed into DINOv2 for feature extraction. If the validation metric does not improve for 10 consecutive epochs, the learning rate is multiplied by 0.9. Regarding the number of Gaussian components, we set it to 4 for the dense point cloud setting and 3 for the GT point cloud setting.

## B. More Quantitative Results

**Per-class Performance Comparison with Existing 3DSGG Methods.** The per-class performance comparisons of GMPSSG against other baselines are presented in Table 8 for objects and Table 9 for predicates.

Notably, benefiting from the effective modeling of intra-class diversity and complex geometric structures by the Gaussian Mixture-distributed Prototypes, our method demonstrates a significant advantage in object recognition. As shown in Table 8, GMPSSG achieves 100% recall on categories with distinct geometric features, such as bath, bed, sink, and toilet. Moreover, compared to MonoSSG, our method achieves a substantial performance leap on flat or structurally complex objects, such as window (59.0→**72.3**) and bookshelf (42.9→**85.7**). Consequently,

*Table 8.* Per-class performance comparison of 3DSGG methods for object recall (%) on 3DSSG (Wald et al., 2020) within 20 node and 8 edge classes, which from ground-truth point cloud. The best and the second best results are highlighted in bold and underlined.

| Method | bath. | bed | bksh. | cab. | chair | cntr. | curt. | desk | door | floor | ofur. | pic. | refri. | show. | sink | sofa | table | toil | wall | wind | mean |
|---|---|---|---|---|---|---|---|---|---|---|---|---|---|---|---|---|---|---|---|---|---|
| IMP | 25.0 | 100 | 28.6 | 47.3 | 67.9 | 64.5 | 74.8 | **58.3** | 47.7 | 96.3 | 17.6 | 58.8 | 11.1 | 14.3 | 53.3 | 65.8 | 61.2 | 92.6 | 83.8 | 54.2 | 56.2 |
| VGfM | 75.0 | 100 | 14.3 | 51.9 | 88.7 | 71.0 | 70.3 | 12.5 | 61.7 | 98.2 | 24.5 | 67.1 | 0 | 14.3 | 71.7 | 71.1 | 72.7 | 88.9 | 85.0 | 51.8 | 59.5 |
| SGPN | 50.0 | 33.3 | 0 | 48.1 | 80.4 | 58.1 | 36.9 | 45.8 | 67.3 | 95.1 | 21.3 | 17.6 | 33.3 | 42.9 | 61.7 | 60.5 | 53.9 | 59.3 | 57.4 | 31.3 | 47.7 |
| SGFN | 50.0 | 33.3 | 0 | 44.2 | 75.4 | 83.9 | 56.8 | 50.0 | 60.7 | 96.9 | 36.2 | 82.4 | 11.1 | 28.6 | 80.0 | 40.8 | 55.2 | 63.0 | 72.0 | 51.8 | 53.6 |
| MonoSSG | 100 | 100 | 42.9 | 62.4 | **92.9** | 80.6 | **89.2** | 33.3 | 81.3 | 98.2 | 57.4 | **96.5** | 44.4 | 100 | 90.0 | **78.9** | 73.9 | 100 | 82.6 | 59.0 | 78.2 |
| Ours | **100** | **100** | **85.7** | **77.5** | 92.1 | **87.1** | 88.3 | 54.2 | 81.3 | **99.4** | **65.4** | 95.3 | **77.8** | 100 | **96.7** | 71.1 | **88.5** | 100 | **85.6** | **72.3** | **85.9** |

*Table 9.* Per-class performance comparison of 3DSGG methods for predicate recall (%) on 3DSSG (Wald et al., 2020) within 20 node and 8 edge classes, which from ground-truth point cloud. The best and the second best results are highlighted in bold and underlined.

| Method | attached to | build in | connected to | hanging on | none | part of | standing on | supported by | mean |
|---|---|---|---|---|---|---|---|---|---|
| IMP | 58.1 | 30.8 | 21.7 | 29.1 | 97.7 | 25.0 | 71.9 | 0 | 41.8 |
| VGfM | 63.1 | 33.3 | 28.3 | 25.4 | **98.1** | 43.8 | 81.5 | 1.1 | 46.8 |
| SGPN | 72.8 | 74.4 | 34.8 | 53.0 | 97.7 | 68.8 | 84.0 | 6.6 | 61.5 |
| SGFN | 68.7 | 66.7 | 56.5 | 57.5 | 96.7 | 75.0 | 81.8 | 2.2 | 63.1 |
| MonoSSG | 74.3 | 87.2 | **65.2** | 56.0 | 97.4 | 37.5 | 92.1 | **8.8** | 64.8 |
| Ours | **77.5** | **89.7** | 54.3 | **70.1** | 97.3 | **100** | **94.5** | 3.3 | **73.4** |

GMPSSG achieves a mean score of **85.9** in object detection, significantly outperforming the runner-up MonoSSG.

Similarly, with regard to the prediction of predicates in Table 9, GMPSSG shows a deep understanding of the semantic relationships within the scene. Our method achieves a remarkable 100% on the part of category, far surpassing MonoSSG, while also maintaining a leading position in relationships such as hanging on and standing on. Although performance fluctuates slightly on a few long-tail categories (*e.g.*, supported by) due to class imbalance, thanks to its robust performance on major relationship classes, GMPSSG elevates the mean predicate accuracy to **73.4**, establishing a new state-of-the-art benchmark.

**Efficiency Analysis.** Table 10 presents the efficiency comparison between GMPSSG, MonoSSG (Wu et al., 2023b), and the Baseline. Notably, to effectively capture intra-class diversity and inter-class similarity, GMPSSG incorporates Gaussian Mixture-distributed Prototypes, which brings the total number of trainable parameters to 11.49M. Despite this increase in model capacity, the memory footprint remains highly efficient, with training memory showing only a marginal increase from 2.06G to 2.14G. Consequently, our method incurs a slight increase in training time compared to the lightweight MonoSSG. Nevertheless, benefiting from its low computational complexity, GMPSSG introduces negligible additional latency during the inference phase compared to the Baseline. Furthermore, given the significant variation in the number of nodes and edges across different scenes, we fixed specific scenes for benchmarking to eliminate bias caused by data scale. Specifically, scene 0a4b8ef6-a83a-21f2-8672-dce34dd0d7ca was used to measure training time and memory, and scene c92fb5ab-f771-2064-842c-c342564aabcc was used for inference speed, thereby ensuring a fair comparison.

We also conducted a more fine-grained runtime analysis. The experiment was performed on scene 0a4b8ef6-a83a-21f2-8672-dce34dd0d7ca, which contains 11 nodes and 21 edges, covering 5 node classes and 3 edge classes. The results in Table 11 show that the image encoder remains the dominant source of training time, taking 828.9 ms. In comparison, our proposed TPI module costs 107.9 ms. Although it introduces some additional overhead, it is not the main bottleneck in the overall training process.

For the Gaussian Mixture-distributed Prototypes related components, the additional cost can be further decomposed as follows: in the node branch, LCL, MDL, and CCL take 55.5 ms, 56.9 ms, and 41.8 ms, respectively, with a total of 154.2 ms; in the edge branch, LCL, MDL, and CCL take 38.3 ms, 42.8 ms, and 37.7 ms, respectively, with a total of 118.8 ms. These results show that MDL, which is the component of main concern, does not dominate the overall additional cost. The MDL costs in the node and edge branches are 56.9 ms and 42.8 ms, respectively, which are on the same order as the other prototype-related losses and are much lower than the computational cost of the image encoder.

**Topology-aware Prototype Interaction Analysis.** We further analyze the effect and limitation of TPI. A strong co-occurrence prior may bias the model against rare but valid interactions, especially for long-tail relation classes. To alleviate this issue, TPI injects the topology prior as a soft knowledge offset rather than hard-coding it or replacing the original prototype parameters. Specifically, it only calibrates prototype means through a gated update, while preserving the covariance and mixture structure. Thus, when relations

*Table 10.* Efficiency comparison on 3DSSG (Wald et al., 2020). We report trainable parameters, training memory, training time, and inference time for each model. See §B for more details.

| Method | Trainable Params↓ | Training Memory↓ | Training Time↓ | Inference Time↓ |
|---|---|---|---|---|
| Baseline | 8.17M | 2.06G | 2s | 0.87s |
| **GMPSSG(Ours)** | 11.49M | 2.14G | 5s | 0.88s |

deviate from co-occurrence statistics, the model can still rely on feature evidence.

Class-level results in the Table 12 further support this observation. With TPI, rare relations such as Connected to and Part of improve from 54.3% to 58.7% and from 18.8% to 25.0%, respectively, although Part of accounts for only 0.8% of the data. Meanwhile, Supported by slightly drops from 17.6% to 15.4%. These results suggest that TPI acts as a mild structural bias rather than simply penalizing low-probability interactions.

## C. More Qualitative Visualization

We provide additional visual comparisons between our method GMPSSG and MonoSSG on **larger-scale scenes** from the 3DSSG dataset. Given the high density of nodes and edges in large-scale scenes, abbreviations are employed for edge predicates to ensure visual clarity. The specific mappings are as follows: **a-attached to**, **b-build in**, **c-connected to**, **h-hanging on**, **n-none**, **p-part of**, **s-standing on**, **u-supported by**.

As illustrated in Fig. 4, we selected three representative scenes (ID: 4a9a43d2-7736-2874-874d-d0fad0570e19, 137a8158-1db5-2cc0-8003-31c12610471e, 7747a4ec-9431-24e8-848f-897279a1e9fe) for visual comparison. The comparative results demonstrate that while MonoSSG performs effectively in general scenarios, our method exhibits significant performance advantages in addressing the prevalent challenges of inter-class similarity and intra-class diversity in 3D indoor scenes.

Mitigating Semantic Ambiguity from Inter-class Similarity: Indoor objects often share similar geometries, causing classification confusion. As shown in the second row, MonoSSG tends to predict coarse-grained labels (*e.g.*, misclassifying specific "cabinet" instances as "otherfurniture"). In contrast, our method leverages enhanced context modeling to capture subtle feature discrepancies, effectively resolving such ambiguity and generating fine-grained labels consistent with the Ground Truth.

Handling Geometric Heterogeneity from Intra-class Diversity: Objects within the same category (*e.g.*, chairs) often exhibit significant morphological variations. In the complex third-row scenario, MonoSSG struggles with such variance, resulting in node misses and erroneous relationships. Conversely, our method demonstrates superior robustness to

deformation and occlusion, accurately identifying diverse instances and reconstructing a complete structure.

## D. Discussion

**Limitations.** One limitation of our method is that the mined Gaussian Mixture-distributed prototypes are still constrained by the distribution of the training data. Therefore, for long-tailed classes with very few samples, the model still struggles to fully capture their intra-class diversity. This is also a common challenge in 3D scene understanding, namely how to learn robust and highly generalizable representations under limited data. To alleviate this issue, in LCL, we adopt a class-conditional negative log-likelihood with the indicator $\mathbb{I}_{i:c}$, so that only samples from class $c$ optimize the corresponding GMP. Meanwhile, compared with a single prototype, multiple Gaussian components provide a more flexible modeling space for limited samples, and TPI further offers complementary structural cues for low-frequency relations. Nevertheless, when some categories contain extremely few training samples, relying only on a more flexible prototype form is still insufficient to fully overcome the limitation caused by data scarcity. In future work, we will explore the use of large-scale models or reward models (Xing et al., 2025b; Qu et al., 2025b;c).

Another limitation is that our validation is currently conducted on indoor scenes. This is mainly because, in 3DSGG, the publicly available benchmark is still largely limited to 3DSSG, which is also consistent with recent literature. Therefore, we follow the standard experimental setting adopted in prior work. To make the evaluation more comprehensive under the current benchmark, we further test the robustness of our method under varying point cloud qualities and a more complex 160/26 label space. Since outdoor 3DSGG benchmarks are still lacking, direct validation on outdoor scenes is currently infeasible. However, our method is designed to address intra-class diversity and inter-class similarity, which may become even more pronounced in outdoor scenes due to more complex spatial relations (Hu et al., 2020). In the future, we plan to build scene graph annotations on existing outdoor 3D segmentation datasets (Liao et al., 2022) for further evaluation.

Moreover, in the current framework, we assign a fixed number of Gaussian components to the Gaussian Mixture-distributed prototypes of each semantic category. However, different categories exhibit varying degrees of geometric and semantic diversity, namely heterogeneous intra-class variations, making a unified fixed configuration potentially suboptimal. To this end, we conducted a preliminary experiment with an adaptive component selection strategy based on class sample proportions (Qu et al., 2025a). In the 3DSSG dataset, the training sample proportions of different categories range from approximately 0.8% to over 15%.

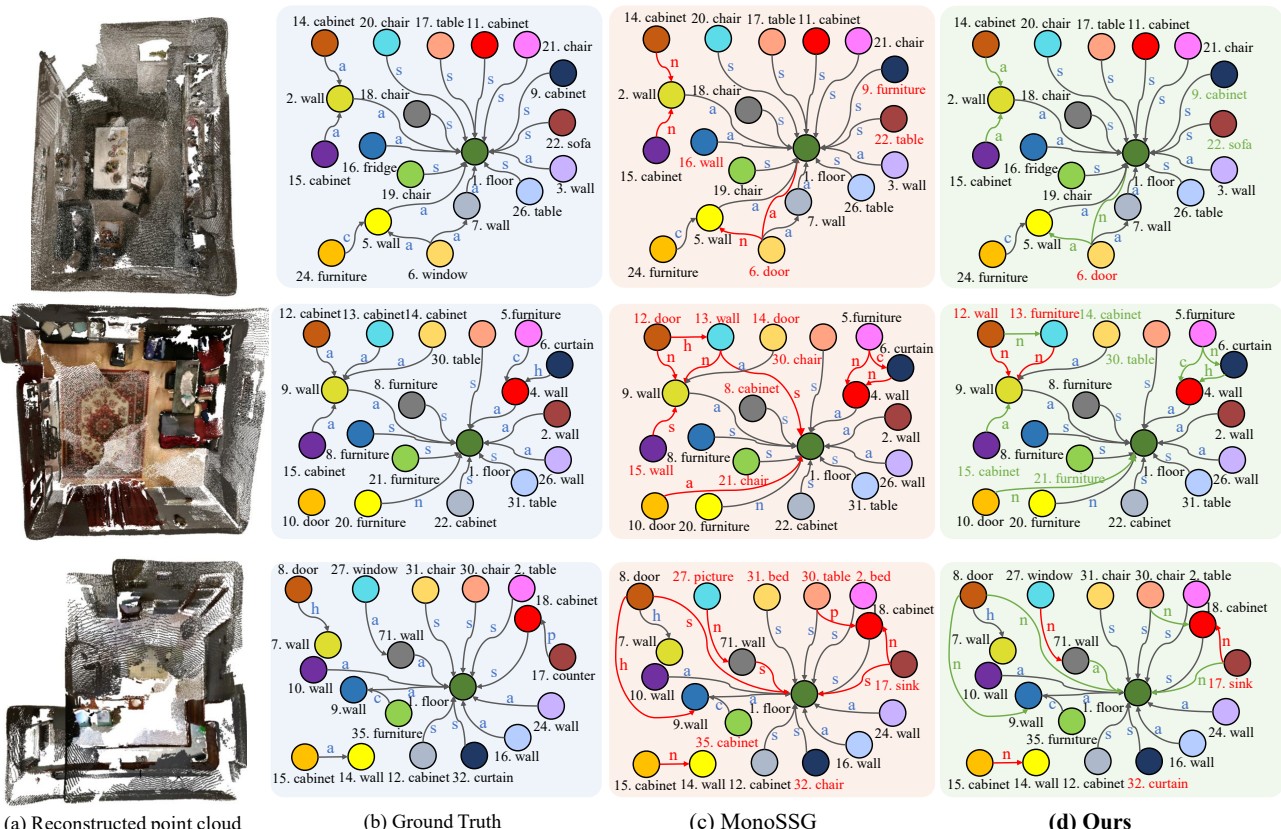

(a) Reconstructed point cloud     (b) Ground Truth     (c) MonoSSG     **(d) Ours**

*Figure 4.* Qualitative results from MonoSSG (Wu et al., 2023b) and our method on the 3DSSG (Wald et al., 2020) dataset. Red edge: misclassified edges from MonoSSG, green edge: edges corrected by our method, red node: misclassified node.

*Table 11.* Detailed training time and trainable parameters of the main modules on 3DSSG (Wald et al., 2020). See §B for more details.

| Module | Image Encoder | TPI | Node_LCL | Node_MDL | Node_CCL | Edge_LCL | Edge_MDL | Edge_CCL |
|---|---|---|---|---|---|---|---|---|
| Training Time | 828.9ms | 107.9ms | 55.5ms | 56.9ms | 41.8ms | 38.3ms | 42.8ms | 37.7ms |
| Trainable Params | 0 | 3.2M | 103.8K | | | 19.0K | | |

*Table 12.* Recall (%) analysis of TPI on five selected predicate classes of 3DSSG (Wald et al., 2020) under the dense point cloud setting. See §B for more details.

| Model | Attached to | Connected to | Part of | Standing on | Supported by |
|---|---|---|---|---|---|
| w/o TPI | 65.4 | 54.3 | 18.8 | 1.0 | 17.6 |
| w/ TPI | 69.6 | 58.7 | 25.0 | 5.0 | 15.4 |

Therefore, we assign $K = 2$, $K = 3$, $K = 4$, and $K = 5$ to categories with sample proportions of 0.8%–5%, 5%–10%, 10%–15%, and over 15%, respectively. The results in Table 13 show that, compared with using a fixed number of components for all categories, this strategy leads to a lower Tri. score but achieves comparable results on most other metrics. Notably, it improves both mRecall metrics, suggesting its potential for alleviating category imbalance. This observation indicates that adaptively determining the number of Gaussian components remains a promising direction for further exploration. Nevertheless, considering the simplicity of the model design, we set the number of Gaussian components for all Gaussian Mixture-distributed prototypes to $K = 4$ in the current version.

**Broader Impact.** This paper proposes GMPSSG, a robust framework for Gaussian Mixture prototype mining. It is designed to overcome the inherent inter-class similarity and intra-class diversity of nodes and edges from the perspective of feature distributions by explicitly modeling their characteristics. The framework incorporates two prototype mining methods tailored for the 3D Scene Graph Generation task: Prototype-anchored Representation Learning and Topology-aware Prototype Interaction. Our method pushes the boundary of 3D scene understanding and holds promise for empowering various real-world applications, such as robot navigation and autonomous localization. However, regarding potential societal impacts, as with common challenges faced by current 3D vision algorithms, our model still exhibits limitations when handling severely occluded or extremely deformed objects. In autonomous systems, such recognition biases may lead to collision risks or erroneous task planning. To mitigate such risks, it is crucial to develop a multi-modal safety verification protocol to ensure system operation (Qu et al., 2026; Xing et al., 2025a).

*Table 13.* Comparison between fixed and adaptive Gaussian component strategies on 3DSSG (Wald et al., 2020) using dense point clouds under the 20-node-class and 8-edge-class setting. See §D for more details.

| $K$ range | Recall(%) | | | | | | mRecall(%) | |
|---|---|---|---|---|---|---|---|---|
| | Tri. ↑ | Tri.*↑ | Obj. ↑ | Obj.* ↑ | Pred. ↑ | Pred.* ↑ | Obj.↑ | Pred.↑ |
| Unique value 4 | 43.0 | 30.6 | 64.8 | 66.4 | 88.3 | 33.2 | 59.3 | 42.0 |
| [2,5] | 40.7 | 31.1 | 64.5 | 66.0 | 84.9 | 33.6 | 60.2 | 43.6 |

---

**Algorithm 1** Pseudo-code of Gaussian Mixture_distributed prototype construction, interaction and self-updating.

---

```python
"""
#_C:_num_classes,_K:_num_components,_D:_embed_dim
#_Node_Head,_Edge_Head:_Instances_of_GMMPrototypeLayer
"""
#======= 1. Prototype Construction (Initialization) =======#
class GMMPrototypeLayer(nn.Module):
    def __init__(self, C, K, D):
        # Initialize learnable GMM parameters (Means, Covariances, Mixing Weights)
        # Parameters are optimized via gradient descent, not EM
        self.mu = Initialize_Params(shape=(C, K, D))
        self.sigma = Initialize_Params(shape=(C, K, D))
        self.pi = Initialize_Params(shape=(C, K))

#======= 2. Topology-aware Prototype Interaction =======#
def prototype_interaction_updating(node_head, edge_head, scene_graph):
    # Fuse components to get class-level prototypes
    P_node = node_head.fuse_components()
    P_edge = edge_head.fuse_components()

    # --- Direction 1: Nodes Contextualize Edges ---
    # Q: Edge Prototypes | K, V: Neighboring Node Prototypes
    Q_edge = select_present_classes(P_edge, scene_graph)
    KV_node = gather_connected_nodes(P_node, scene_graph)

    # Update Edge Prototypes via Cross-Attention
    Context_edge = MultiHeadAttn(Q=Q_edge, K=KV_node, V=KV_node)
    P_edge_new = GatedUpdate(Q_edge, Context_edge)

    # Hard update: Apply refined prototypes to learnable parameters
    edge_head.update_means(P_edge_new)

    # --- Direction 2: Edges Contextualize Nodes ---
    # Q: Node Prototypes | K, V: Neighboring Edge Prototypes
    Q_node = select_present_classes(P_node, scene_graph)
    KV_edge = gather_connected_edges(P_edge, scene_graph)

    Context_node = MultiHeadAttn(Q=Q_node, K=KV_edge, V=KV_edge)
    P_node_new = GatedUpdate(Q_node, Context_node)

    node_head.update_means(P_node_new)

#======= 3. Prototype-anchored Representation Learning =======#
def prototype_self_updating(head, x, labels, other_protos):
    # A.Likelihood-driven Clustering Learning
    # Compute log-likelihood N(x|mu, sigma) and soft assignment
    L_cls = head.compute_classification_loss(x, labels)

    # B. Cohesion-oriented Constraint Learning.
    L_reg = head.compute_geometric_reg(self.mu, self.sigma)

    # C. Mutual Information-guided Separation Learning.
    P_self = head.fuse_components()
    L_con = symmetric_kl_contrast(P_self, other_protos)
```

