# OpenReview forum: "Learning Gaussian Mixture-distributed Prototypes for 3D Scene Graph Generation from RGB-D Sequences"
_ICML.cc/2026/Conference — ICML 2026 regular_

### Official Review · Reviewer_uPm2 · 2026-03-02

**Soundness:** 2
**Presentation:** 2
**Significance:** 2
**Originality:** 2
**Overall Recommendation:** 4
**Confidence:** 2

**Summary:**

This paper proposes GMPSSG for 3D Scene Graph Generation (3DSGG) from RGB-D sequences. Each node and edge category is modeled as a Gaussian Mixture-distributed Prototype to address intra-class diversity and inter-class similarity. Two learning strategies are introduced: (1) Prototype-anchored Representation Learning with three sub-losses for constructing a structured category space; (2) Topology-aware Prototype Interaction for calibrating prototypes via cross-attention and gating. Experiments on 3DSSG show improvements over existing methods (Tables 1–2).

**Compliance With Llm Reviewing Policy:**

Affirmed.

**Final Justification:**

My concerns have been addressed, and I will keep the positive score. However, as mentioned in my original review, I am not an expert in this field.

**Key Questions For Authors:**

**Q1.** How is the symmetric KL divergence between GMMs (Eq.13–14) computed in practice? Please specify the approximation method and its complexity. **This directly affects reproducibility assessment.**


**Q2.** In Table 6, "Our" improves over Sinkhorn EM by only 1.6% Tri. Is this gain from a specific PRL sub-loss or from TPI? A finer-grained ablation (Standard EM + each sub-loss incrementally) would clarify the contribution source.

As I am not an expert in this field, I may need to refer to other reviewers' comments to determine the final score.

**Limitations:**

Yes.

**Strengths And Weaknesses:**

## Strengths

**1. Well-motivated problem formulation (Originality).**

Approaching 3DSGG from a feature distribution perspective is a meaningful departure from single-prototype methods (Wu et al., 2023b; Feng et al., 2025a).

**2. Systematic ablation studies (Soundness).**

Tables 3–7  clearly validate each component: GMP+PRL yields +7.2% Tri. (Table 3); three sub-losses show incremental gains (Table 4); component number, update strategies, and interaction directions are each explored (Tables 5–7).

**3. Strong gains in the large-scale setting (Significance).**

Table 2 (160 nodes, 26 edges) shows mObj +8.4% and mPred +4.9%, demonstrating that the method scales well when inter-class confusion intensifies with more categories.

---

## Weaknesses

**1. Limited technical novelty (Originality).**

GMM-based category prototypes have been explored in GMMSeg. The three PRL sub-losses each have direct precedents: Eq.12 is standard GMM negative log-likelihood; Eq.14 is symmetric KL-based contrastive regularization; Eq.15 is a simple mean-distance and variance penalty. The paper does not sufficiently articulate the **technical** (not application-level) distinction from GMMSeg.

**2. Key technical details missing or questionable (Soundness).**

Eq.13–14 use symmetric KL divergence between GMMs, which has no closed-form solution. No approximation method is specified anywhere, hindering reproducibility.

**3. Narrow evaluation scope (Significance).**

All experiments use only the 3DSSG dataset. No other benchmarks are tested despite the generalizability claims in paper.

---

> ### Author Rebuttal · Authors · 2026-03-31
>
> We thank reviewer uPm2 for the constructive feedback. We provide point-to-point response below.
>
> ----
>
> **Q1: Technical novelty of GMPSSG**
>
> **A1:** Sorry for the confusion. Our contribution is not to invent NLL or KL formulas, but to unify them, together with TPI for co-occurrence priors, into a Gaussian Mixture Prototype (GMP) mining network for 3DSGG. Thus, our method differs from GMMSeg not only in application, but also in modeling objectives and optimization.
>
> First, modeling objective differs. GMMSeg uses GMMs as dense generative classifiers in place of softmax. In contrast, we use class-wise GMMs as explicit class prototypes, shifting the focus from generative classification to prototype-level modeling of intra-class diversity and inter-class similarity.
>
> Second, optimization differs. GMMSeg relies on online Sinkhorn-EM, which ignores inter-class overlap and is prone to collapse. By contrast, our PRL enables stable prototype learning. Eq.12 uses an indicator function to prevent head classes from dominating tail classes, ensuring stable updates even with sparse samples. Eq.14 adds a structural loss absent in GMMSeg. Because few features limit instance-level contrast, we use symmetric KL divergence at the distribution level. MDL further improves distribution-level discrimination through efficient node/edge negative sampling and by fusing Gaussians into class-level diagonal prototypes. Eq.15 complements Eq.14 by directly regularizing prototypes to improve compactness and decorrelation. Together, the three terms promote intra-class aggregation, inter-class decorrelation, and compactness within one prototype learning network.
>
> Finally, motivated by the properties of 3DSGG, we propose TPI, which is absent from both GMMSeg and prior 3DSGG methods. TPI captures scene co-occurrence priors and calibrates node and edge prototype distributions through dynamic updates to improve node-edge matching.
>
> In summary, motivated by the challenges of inter-class similarity and intra-class diversity, our technical contribution lies in combining PRL and TPI into a GMP mining network tailored for 3DSGG. This makes our method distinct from GMMSeg and prior 3DSGG methods in modeling objectives, optimization, and use of task-specific structure.
>
> **Q2: Details for symmetric KL.**
>
> **A2:** Thanks for pointing this out. We do not directly compute the symmetric KL divergence between GMPs. Instead, for each class, we fuse its Gaussian components into a class-level diagonal Gaussian via moment matching, using mixture-weighted first and second moments. We then compute the symmetric KL divergence between these class-level diagonal Gaussians, which has a closed-form expression. This also reduces computational cost. The resulting distance is then used to encourage inter-class separability. We will add these details to Sec.3.5.
>
> **Q3: Validated only in 3DSSG.**
>
> **A3:** Thanks for your comment. We validate only on 3DSSG because it remains the main public benchmark for 3DSGG, consistent with recent work. For example, HEDSGP states that 3DSSG is the only public dataset for this task. We therefore follow prior work. Still, we broaden evaluation by testing robustness under different point cloud qualities and a more complex 160/26 label space. The sentence "These results validate the generalization of our method" may indeed sound too strong. In Sec.4.2, it only summarizes results under different input qualities and a larger label space in the same benchmark, rather than cross-dataset generalization. We apologize for the ambiguity and will revise the text.
>
> **Q4: Details for symmetric KL.**
>
> **A4:** Thanks for your suggestion. Please see Q2.
>
> **Q5: Contributions of TPI and PRL.**
>
> **A5:** Thanks for your suggestion. Tables 3, 4, and 7 show the contributions of PRL and TPI. In Table 3, adding PRL raises Tri. to 42.2, and TPI further raises it to 43.0, showing that PRL provides the main gain and TPI adds further improvement. **Table 4 shows cumulative effects of the three PRL losses.** With only $L_{LCL}$, Tri. reaches 41.2. Adding $L_{MDL}$ and $L_{CCL}$ raises it to 42.2, showing that $L_{LCL}$ is the main contributor, while the other two further refine the prototype space. Table 7 shows the effect of TPI’s interaction design, where comparison with one-way interaction verifies the effectiveness of the two-way strategy.
>
> We also conduct a standard EM ablation. However, standard EM and Sinkhorn EM rely on closed-form online EM updates with momentum, whereas PRL uses end-to-end gradient optimization. Updating all parameters by gradients changes the original EM logic, so the results are not comparable to ours. Moreover, standard EM plays a role similar to LCL, so we exclude it. We add only $L_{CCL}$, $L_{MDL}$, and TPI, and the results are as follows:
> |Standard EM|$L_{CCL}+L_{MDL}$|TPI|Tri.|Obj.|Pred.|mRecall Obj.|mRecall Pred.|
> |:-|:-|:-|:-|:-|:-|:-|:-|
> |√|||41.1|62.9|87.8|58.2|40.6|
> |√|√||41.8|63.5|88.7|58.5|39.4|
> |√|√|√|42.1|63.9|89.8|58.3|37.2|

---

> > ### Author Rebuttal · Reviewer_uPm2 · 2026-04-04
> >
> > Thank you for the detailed rebuttal. My concerns have been addressed, and I will keep the positive score.
> >
> > However, as mentioned in my original review, I am not an expert in this field. I would suggest the authors further improve the paper by addressing the comments from other reviewers.

---

> > > ### Author Response · Authors · 2026-04-05
> > >
> > > Dear Reviewer uPm2,
> > >
> > > Thank you for your response. We are very glad that our rebuttal has addressed your concerns.
> > >
> > > We deeply value the constructive comments from all reviewers, and we will carefully consider and address each piece of feedback to further improve the quality of our manuscript.
> > >
> > > Thank you again for your support.
> > >
> > > Best regards,
> > >
> > > Authors

---

### Official Review · Reviewer_UwCW · 2026-03-12

**Soundness:** 4
**Presentation:** 3
**Significance:** 2
**Originality:** 3
**Overall Recommendation:** 4
**Confidence:** 3

**Summary:**

This paper focus on intra-class diversity and inter-class similarity of nodes/edges in 3DSGG, proposing the GMPSSG framework with Gaussian mixture prototypes for feature distribution modeling and two core learning mechanisms for prototype optimization. Experiments show it outperforms SOTA, providing new insights for 3D scene feature distribution modeling.

**Compliance With Llm Reviewing Policy:**

Affirmed.

**Final Justification:**

Thank you for your rebuttal. I maintain my positive score.

**Key Questions For Authors:**

1. Have you tried designing adaptive Gaussian component numbers for different categories, and are there any preliminary experimental results?

**Limitations:**

yes

**Strengths And Weaknesses:**

Strength:

（1）Addresses 3DSGG from the feature distribution perspective, using GMP to explicitly characterize intra-class diversity and inter-class similarity of nodes/edges, breaking single-prototype limits and capturing real feature distributions accurately.

（2）Prototype-anchored Representation Learning shapes prototypes in three dimensions, builds an independent and well-structured category space, enhances prototype discriminability and mines node/edge feature distributions deeply.

（3）Topology-aware Prototype Interaction captures implicit co-occurrence priors in 3D scenes, dynamically calibrates prototype distribution, and ensures plausible node-edge matching consistent with scene topology.



Weakness:

1. The number of Gaussian components is fixed, failing to adapt to diverse categories and causing modeling redundancy or insufficiency.
2. Weak in modeling long-tail classes due to training data distribution limits, unable to capture their intra-class diversity.
3. Validated only on indoor 3D scenes without outdoor tests, so its generalization needs verification.

---

> ### Author Rebuttal · Authors · 2026-03-31
>
> We thank reviewer UwCW for the valuable time and constructive feedback. We provide point-to-point response below.
>
> ----
>
> **Q1: Number of Gaussian components for each category.**
>
> **A1:** Thanks for pointing out this promising direction. Since different classes exhibit different degrees of geometric and semantic heterogeneity, we conduct an additional experiment in which we introduce a data statistics stage at the beginning of training. This stage automatically determines the number of Gaussian components K for each class based on its proportion of training samples. In the 3DSSG dataset, the proportion of training samples for each class ranges from 0.8% to more than 15%. Accordingly, we assign K=2 to classes with 0.8%-5% of the training samples, K=3 to classes with 5%-10%, K=4 to classes with 10%-15%, and K=5 to classes with more than 15%. The results show that, compared with using a fixed value for all prototypes, this strategy performs worse than our method on the Tri metric, while it remains largely comparable on the other metrics. Interestingly, it outperforms our method on both mRecall metrics, which suggests better control of class balance. This observation indicates the potential of adaptively determining the number of Gaussian components. However, for simplicity, in the current version we directly set the number of Gaussian components for all Gaussian mixture prototypes (GMP) to K=4.
>
> |K range|Tri.|Tri.* |Obj.|Obj.*|Pred.|Pred.*|mRecall Obj.|mRecall Pred.|
> |:-|:-|:-|:-|:-|:-|:-|:-|:-|
> |unique value 4|43.0|30.6|64.8|66.4|88.3|33.2|59.3|42.0|
> |[2,5]|40.7|31.1|64.5|66.0|84.9|33.6|60.2|43.6|
>
> As you note, automatically determining k for different classes is an interesting problem. We find that some prior works, such as [ref1], study automatic selection of the number of Gaussian components. However, they only consider one-dimensional GMMs. After running their code, we find that these methods are complex and time-consuming. More importantly, adapting them properly to 3D scene graph generation is still challenging because 3DSGG has a severe long-tail distribution. We therefore leave this direction as future work and will add this discussion to Appendix Sec.D.
>
> [ref1] Robust Model Selection and Nearly-Proper Learning for GMMs. NeurIPS 2022.
>
> **Q2: Diversity modeling for long-tail classes.**
>
> **A2:** Thanks for pointing this out. Since long-tail classes have limited training samples, their intra-class diversity is harder to capture well. This is a common challenge in 3DSGG.
>
> Our method includes several design choices to ease this issue. Specifically, in LCL, we use a class-conditional negative log-likelihood with the indicator $\mathbb{I}_{i:c}$, so that only samples from class $c$ optimize the corresponding GMP, which prevents features of scarce classes from being biased toward head classes during optimization and ensures that long-tail classes are still modeled. In addition, compared with a single prototype, multiple Gaussian components allow the model to assign different few samples to different components, which helps capture diversity even when data are limited. TPI further provides complementary structural cues for low-frequency relations.
>
> That said, we agree that when a class has very few training samples, even a more flexible prototype form cannot overcome the limitation imposed by the data distribution, and its diversity still cannot be well modeled. In future work, we will further explore strategies such as data augmentation and memory banks to ease this issue. We will add this discussion to Appendix Sec.D.
>
> **Q3: Validated without outdoor tests.**
>
> **A3:** Thanks for pointing this out. We validate only on indoor scenes because, in 3DSGG, the publicly available benchmark is still mainly 3DSSG. This is also consistent with recent literature. For example, HEDSGP explicitly states that 3DSSG is the only publicly available dataset for this task. We therefore follow the standard experimental setting in prior work. Still, we maximize current benchmark evaluations by testing robustness against varying point cloud qualities and a complex 160/26 label space. As outdoor benchmarks are lacking, direct validation is currently infeasible. However, our method is designed to address intra-class diversity and inter-class similarity. In outdoor scenes, these issues are more pronounced because spatial relations are often more complex [ref2], which suggests that our method may also be useful. In the future, we plan to build scene graph annotations on existing outdoor 3D segmentation datasets [ref3] for future evaluation. We will add this discussion to Appendix Sec.D.
>
> [ref2] Randla-net: Efficient semantic segmentation of large-scale point clouds CVPR 2020
>
> [ref3] Kitti-360: A novel dataset and benchmarks for urban scene understanding in 2d and 3d TPAMI 2022
>
> **Q4: The experiment about number of Gaussian components for each category.**
>
> **A4:** Thank you for your suggestion. You can see the Q1.

---

> > ### Author Rebuttal · Reviewer_UwCW · 2026-04-04
> >
> > I thank the authors for their thorough explanations and the supplementary experiments provided in the rebuttal. The preliminary results regarding the adaptive number of Gaussian components are particularly insightful; while the Triplet metric slightly decreased, the improvement in mRecall highlights the potential of an adaptive mechanism to mitigate class imbalance, which is critical for addressing the severe long-tail distribution inherent in 3D scene graph generation. Additionally, the clarifications regarding the modeling logic for long-tail categories and the current limitations of outdoor benchmarks are reasonable. I support the decision to include these discussions and future research directions in the Appendix.

---

> > > ### Author Response · Authors · 2026-04-05
> > >
> > > Dear Reviewer UwCW,
> > >
> > > Thank you for your positive feedback and for supporting our work. We are very glad that our explanations and the supplementary experiments successfully addressed your concerns. We will add the corresponding discussion in the final version.
> > >
> > > Thank you again for your constructive suggestions.
> > >
> > > Best regards,
> > >
> > > Authors

---

### Official Review · Reviewer_ErV5 · 2026-03-13

**Soundness:** 2
**Presentation:** 3
**Significance:** 2
**Originality:** 2
**Overall Recommendation:** 3
**Confidence:** 3

**Summary:**

The paper addresses the critical challenges of intra-class diversity and inter-class similarity in 3D Scene Graph Generation (3DSGG) from RGB-D sequences. It presents GMPSSG, a framework that replaces traditional single-vector prototypes with Gaussian Mixture-distributed Prototypes (GMP) to better model the complex feature distributions of objects and relations. The framework incorporates two primary learning modules: Prototype-anchored Representation Learning (PRL), which uses likelihood-driven clustering, mutual information-guided decorrelation and cohesion-oriented regularization to structure the category space, and Topology-aware Prototype Interaction (TPI), which leverages co-occurrence priors to calibrate prototype distributions through cross-attention mechanisms. Experiments on the 3DSSG dataset demonstrate the effectiveness of GMPSSG.

**Compliance With Llm Reviewing Policy:**

Affirmed.

**Key Questions For Authors:**

1)	The authors employ a fixed number of Gaussian components for every semantic category. Table 5 shows that the performance is somewhat sensitive to K. As different categories exhibit varying degrees of geometric and semantic heterogeneity, will applying the same number of components to every category lead to over-parameterization for simpler classes and under-fitting for complex ones? Will adaptively determining K per class be a better solution?
2)	The Topology-aware Prototype Interaction (TPI) module injects co-occurrence priors into the prototype means to ensure topological plausibility. As the authors state the model still struggles with long-tailed classes, is it possible that TPI aggravates this by penalizing valid but statistically ‘unlikely’ interactions, potentially leading to poor generalization in unconventional environments and rare objects?
3)	Initialization of GMM parameters strongly affects convergence and collapse risk. While Algorithm 1 mentions ‘Initialize learnable GMM parameters’, the manuscript lacks a specific description of the method used. Providing these details is essential for reproducibility.
4)	While Table 10 provides a brief efficiency comparison, the paper does not clearly analyze which components of the proposed framework (e.g., MDL, TPI, or other modules) dominate the computational overhead. A more detailed breakdown of the runtime cost across different components would help to address concerns regarding whether pairwise KL-divergence computations across class distributions in the MDL module will restrict the application of this method to datasets with a larger number of categories.
5)	Some typos are found. For example:
a)	Page 1, Fig 1, ‘handing on’ should be ‘hanging on’;
b)	Page 8, Table 6, ‘Updata’ should be ‘Update’;
c)	Page 8 & 13, ‘miss-classified’ should be ‘misclassified’.

**Limitations:**

Yes, in the appendix part.

**Strengths And Weaknesses:**

Strengths:
1. The idea is straightforward, and performace improvement proves the soundness of the idea.
2. The overall writing is easy to follow.

Weakness:
1. Some details need more clarification, please check the questions part.
2. The comparison with existing methods misses some work in top conferences after 2024.

---

> ### Author Rebuttal · Authors · 2026-03-31
>
> We thank reviewer ErV5 for the constructive feedback and provide our point-to-point responses below, which will be added to the relevant sections.
>
> ----
>
> **Q1: The comparison with existing methods.**
>
> **A1:**  Our work focuses on 3DSGG from RGB-D images, and Table 1 includes **SCRSSG (ICCV 2025)** and **HEDSGP (TPAMI 2025)**. Comparison with point-cloud-based methods is difficult due to differences in preprocessing and evaluation. Even so, we adapt and compare methods including VL-SAT. We also adapt **OCRL** [ref1] for broader comparison. We will add the results to Table 1.
> |Method|Tri.|Tri.*|Obj.|Obj.*|Pred.|Pred.*|mRecall Obj.|mRecall Pred.|
> |:-|:-|:-|:-|:-|:-|:-|:-|:-|
> |OCRL|30.0|23.7|52.5|53.9|88.6|26.4|40.4|27.9|
> |Ours|43.0|30.6|64.8|66.4|88.3|33.2|59.3|42.0|
>
> [ref1] Object-Centric Representation Learning for Enhanced 3D Semantic Scene Graph Prediction. NeurIPS 2025.
>
> **Q2: Number of Gaussian components for each category.**
>
> **A2:** Since various classes exhibit various degrees of geometric and semantic heterogeneity, we perform a statistics stage at the start of training to set the number of Gaussian components K for each class based on its proportion in the training set. In 3DSSG, the proportion ranges from 0.8% to over 15%. We set K=2 for classes with 0.8%-5% of traing samples, K=3 for 5%-10%, K=4 for 10%-15%, and K=5 for over 15%. Compared with a fixed K for all classes, this strategy performs worse on Tri., but better on both mRecall metrics, indicating that it achieves better class balance. This shows the potential of adaptive K. For simplicity, the current version uses a fixed K=4.
> |K range|Tri.|Tri.* |Obj.|Obj.*|Pred.|Pred.*|mRecall Obj.|mRecall Pred.|
> |:-|:-|:-|:-|:-|:-|:-|:-|:-|
> |4|43.0|30.6|64.8|66.4|88.3|33.2|59.3|42.0|
> |[2,5]|40.7|31.1|64.5|66.0|84.9|33.6|60.2|43.6|
>
> As you note, automatically determining K for each class is interesting. Some prior works, such as [ref2], study automatic selection of the number of Gaussian components, but only for one-dimensional GMMs. After running their code, we found the method complex and time-consuming. We therefore leave this to future work.
>
> [ref2] Robust Model Selection and Nearly-Proper Learning for GMMs. NeurIPS 2022.
>
> **Q3: Limitations of TPI.**
>
> **A3:** We agree that if the co-occurrence prior is too strong, it may bias the model against interactions that are rare but valid, especially for long-tail classes.
>
> We mitigate this in TPI by encoding the prior as a knowledge offset and injecting it. TPI neither hard-codes the topology prior nor replaces the original parameters. Instead, it updates prototype means through a gated update. When a relation deviates from co-occurrence statistics, the learned prototypes can still rely on feature evidence. And TPI only calibrates the means, which helps preserve flexibility.
>
> We also report class-level results. After adding TPI, rare relations benefit: Connected to improves from 54.3% to 58.7%, and part of improves from 18.8% to 25.0%, which accounts for only 0.8%. Some classes drop, such as Supported by, from 17.6% to 15.4%. These results suggest that TPI does not simply penalize low-probability interactions, but serves as a mild structural bias. But this remains a challenge in unconventional environments and rare objects.
> |Model|Attached to|Connected to|Part of|Standing on|Supported by|
> |:-|:-|:-|:-|:-|:-|
> |w/o TPI|65.4%|54.3%|18.8%|1%|17.6%|
> |w TPI|69.6%|58.7%|25.0%|5%|15.4%|
>
> **Q4: Initialization of GMM parameters.**
>
> **A4:** The means of GMP are initialized by a truncated normal with $\sigma=0.02$, introducing differences across components. The variances are initialized to 0 and passed through a softplus with lower bound $\sigma_{\min}=0.03$. The mixture weights are learned from logits initialized to 0, and softmax yields uniform initial weights.
>
> **Q5: Detailed breakdown of runtime cost.**
>
> **A5:** We analyze runtime on a scene with 11 nodes, 21 edges, 5 node classes, and 3 edge classes. The image encoder costs 828.9 ms, while TPI takes 107.9 ms. For GMP, node branch costs are 55.5 ms for LCL, 56.9 ms for MDL, and 41.8 ms for CCL; while edge branch costs are 38.3 ms, 42.8 ms, and 37.7 ms. Thus, MDL does not dominate the cost. We also test the 160/26 setting, where the scene contains 30 nodes, 147 edges, 15 node classes, and 13 edge classes. MDL takes 163.5 ms and 81.6 ms for node and edge. This shows that MDL cost grows with class and feature numbers, but not exponentially. This is because moment matching fuses Gaussians into one diagonal prototype per class before symmetric KL computation. Notably, PRL adds no inference cost because it is discarded.
> |Module|Image encoder|TPI|Node_LCL|Node_MDL|Node_CCL|Edge_LCL|Edge_MDL|Edge_CCL|
> |:-|:-|:-|:-|:-|:-|:-|:-|:-|
> |Training Time(ms)|828.9|107.9|55.5|56.9|41.8|38.3|42.8|37.7|
> |Trainable Params|0|3.2M||103.8k|||19.0k|
>
> **Q6: Some typos about paper.**
>
> **A6:** Thanks for your careful review. We correct all mentioned typos and carefully proofread the paper.

---

### Decision · Program_Chairs · 2026-04-30

**Decision:**

Accept (regular)

**Comment:**

The scores by the reviewers are: 1x WR, 2x WA, however the negative reviewer did not acknowledge the rebuttal.

For the AC, the contribution is somewhat limited, as it focuses on a specific problem in GMPSSG (interclass similarity), but the authors did provide satisfying answers to the concerns raised by the reviewers. However, the authors should clarify better the contribution in the introduction of the main paper in a revision.